# Multiple Kernel Stein Spatial Patterns for the Multiclass Discrimination of Motor Imagery Tasks

**Steven Galindo-Noreña** * , **David Cárdenas-Peña** and **Álvaro Orozco-Gutierrez**

Automatics Research Group, Universidad Tecnológica de Pereira, 660003 Pereira, Colombia; dcardenasp@utp.edu.co (D.C.-P.); aaog@utp.edu.co (Á.O.-G.)
* Correspondence: sgalindo@utp.edu.co; Tel.: +57-6-313-7300 (ext. 7680)

**Abstract:** Brain–computer interface (BCI) systems communicate the human brain and computers by converting electrical activity into commands to use external devices. Such kind of system has become an alternative for interaction with the environment for people suffering from motor disabilities through the motor imagery (MI) paradigm. Despite being the most widespread, electroencephalography (EEG)-based MI systems are highly sensitive to noise and artifacts. Further, spatially close brain activity sources and variability among subjects hampers the system performance. This work proposes a methodology for the classification of EEG signals, termed Multiple Kernel Stein Spatial Patterns (MKSSP) dealing with noise, raveled brain activity, and subject variability issues. Firstly, a bank of bandpass filters decomposes brain activity into spectrally independent multichannel signals. Then, Multi-Kernel Stein Spatial Patterns (MKSSP) maps each signal into low-dimensional covariance matrices preserving the nonlinear channel relationships. The Stein kernel provides a parameterized similarity metric for covariance matrices that belong to a Riemannian manifold. Lastly, the multiple kernel learning assembles the similarities from each spectral decomposition into a single kernel matrix that feeds the classifier. Experimental evaluations in the well-known four-class MI dataset 2a BCI competition IV proves that the methodology significantly improves state-of-the-art approaches. Further, the proposal is interpretable in terms of data distribution, spectral relevance, and spatial patterns. Such interpretability demonstrates that MKSSP encodes features from different spectral bands into a single representation improving the discrimination of mental tasks.

**Keywords:** brain–computer interface; motor imagery; electroencephalography; multiple kernel learning

## 1. Introduction

Brain–computer interfaces (BCI) are systems that aim to establish a direct connection between the human brain and a computer. These systems can convert specific patterns of brain electrical activity from electroencephalographic (EEG) signals into commands to control external devices, allowing direct interaction with the environment [1]. Thanks to the provided alternative means of mobility and communication, BCI systems become especially useful in people suffering from motor disabilities such as amyotrophic lateral sclerosis, cerebral strokes, and paralysis, [2,3]. In particular, the motor imagery (MI) paradigm takes advantage of the mental simulation of actions without physical execution to capture electrical activity patterns from the sensory-motor cortex [4]. BCI systems translate the MI-associated task pattern from electroencephalography (EEG) into control commands.

Despite the potential of EEG-based systems, the background noise, muscle artifacts, head movements, and blinks largely detriment its performance, which constrains its introduction into real-world applications [5,6]. One way to deal with the above issues comprises the design of robust feature sets. Among the wide variety of feature extraction approaches, the Common Spatial Patterns (CSP) stands out for capturing the MI activity within the EEG. CSP projects signals from two classes



into a space with maximum variance for one class and minimum for the other, providing discriminant features [7]. In this sense, the relative power in the CSP space, computed from the channel-wise covariance matrix, characterizes a band-passed multichannel EEG trial. Nonetheless, the interclass variability within subjects hampers the system effectiveness for MI applications [8].

As an initial solution, multiple CSP-based variants optimize the spectral filtering for yielding discriminative features under varying conditions. For instance, the Common Spatio-Spectral Patterns (CSSP) approach enhances a finite impulse response (FIR) filter by incorporating a time delay for filtering to improve CSP performance proposed [9]. The Common Sparse Spectral-Spatial Patterns modified the CSSP by simultaneously tuning the FIR and CSP filters [10]. Later, Novi et al. introduced the CSP features from multiple sub-bands for EEG classification with a score fusion strategy [11]. In the same approach, the Filter-Bank CSP (FBCSP) exploited the potential correlation between CSP characteristics extracted from different bands to improve the signal discrimination [8]. In general, filter-banked feature extraction approaches outperform conventional multichannel time-series representations, including traditional CSP, in supervised learning schemes [12]. Nonetheless, spectral variants of CSP hardly decode MI tasks that activate spatially close regions, no matter the frequency differences [13].

A different solution decodes the trial variability through Riemannian manifolds composed of Symmetric Positive Definite (SPD) matrices, i.e., trial covariance estimations. Due to preserving the geometric structure and behaving like a matrix Hilbert space, the Riemannian manifolds devote conventional pattern recognition machines to time-series classification [14]. However, the covariance matrices as features suffer from the curse of dimensionality, since the usual matrix dimensions compare to the number of training samples [15]. For coping with high-dimensionality, various approaches map data from the Riemannian manifold into lower-dimensional vector spaces. For instance, the extension of three nonlinear dimension reduction (DR) approaches, namely, Local Linear Embedding (LLE), Hessian LLE, and Laplacian Eigenmaps, to the Riemannian geometry allowed the motion and image segmentation from a clustering point of view [16]. Nonetheless, such nonlinear DR algorithms lack a parametric mapping to the low-dimensional space, depending on an interpolation stage [17]. In addition, Principal Geodesic Analysis (PGA) emerges as a principal component analysis generalization for Riemannian manifolds by finding a tangent space with maximized variance [18]. Two subsequent tangent space approaches introduce kernel-based mappings [16,19]. However, tangent space projections distort data structure, with larger distortions at regions far from the space origin [20]. Another DR alternative maps from high to low-dimensional manifolds, where the resulting output manifold serves as input for existing SPD-based algorithms [15]. As an example, a linear mapping takes advantage of provided labels to maximize the geodesic distance among samples from different classes while minimizing distances among equally-labeled samples [17]. Despite favoring the supervised tasks, the linear combination of distances as a cost function underperforms at inherently nonlinear distributed classes.

For dealing with the above issues, this work proposes a methodology for the classification of SPD matrices with three main contributions: firstly, we introduce a dimension reduction for SPD matrices. Contrarily to other approaches, the approach uses the provided labels to learn the mapping that not only preserves the data structure but also favors the class discrimination thanks to a supervised learning scheme. Secondly, we propose two kernel mappings to decode the nonlinear relationships that enhance the separability of classes at the component level, namely, the Stein kernel devoted to SPD matrices, and the linear combination of kernel functions unifying the knowledge from different spectral bands.

The methodology, termed Multi-Kernel Stein Spatial Patterns (MKSSP), includes a spectral decomposition, spatial filtering, and supervised kernel learning stages to enhance the discrimination of multichannel EEG signals. Firstly, a bank of bandpass filters decomposes the multichannel signals, from which MKSSP computes the covariance matrices. Then, the spatial filtering projects the band-wise covariances into smaller-dimensional Riemannian manifolds by maximizing a supervised kernel-based

cost function. The Jensen-Bregman LogDet Divergence, devoted to SPD matrices, parameterizes the considered kernel pair-wise comparing EEG trial at each band. Finally, a multiple-kernel yields a single similarity measure that feeds a classifier. The centered kernel alignment (CKA) works as the cost function for training the dimension reduction and the multikernel. We tested our methodology in the Dataset IIa of the BCI Competition IV, outperforming the state of the art in the four task MI classification for most of the subjects and achieving significant improvements for the ones with usual low-performance scores.

Aiming to gain insight into the methodology benefits, we analyze the methodology from four perspectives. The data analysis evidences the class separation. The noise sensitivity validates the robustness and class unbiasing. The spectral and spatial points of view explain the subject variability and model interpretability, respectively.

The paper outline is as follows. Section 2 describes the EEG decomposition, the time-series similarity through the Stein kernel, the proposed spatial filter optimization and multiple kernel learning based on CKA. Section 3 presents the evaluation setup for the proposed MKSSP. Section 4 describes the classification results, the comparison with other methods, and the model interpretability. Lastly, Section 5 concludes the manuscript discussing the main findings.

## 2. Methods

### 2.1. EEG Decomposition

Let a set of $N$ labeled multichannel EEG time series (trials), acquired from a single subject $\mathcal{X} = \left\{ x_n(t) \in \mathbb{R}^C, y_n \in \mathcal{L} \right\}_{n=1}^N$, where $C$ stands for the number of channels, $t \in [1, T]$ indexes the time instants, and $y_n$ labels the $n$-th time series $x_n(t)$. $\mathcal{L}$ defines the set of possible classes, usually related to mental states.

To take advantage of the spectral content of the EEG signals, each trial is band-passed through a set of $B$ filters, achieving the filter-banked time series representation of EEG data described in Equation (1):

$$\mathcal{X}_B = \left\{ x_{nb}(t) = [x_{nc}(t) * h_b(t)]_{c=1}^C, b \in [1, B] \right\}_{n=1}^N \tag{1}$$

where $x_{nc}(t) \in \mathbb{R}$ stands for the $c$-th channel from the $n$-th trial, $h_b(t) \in \mathbb{R}$ corresponds to the impulse response of the $b$-th linear phase FIR filter, and $*$ denotes the convolution operator. Afterwards, a band-wise spatial filtering linearly mixes the input channels into components at each time instant resulting in a new set of band-filtered time series as follows:

$$\mathcal{X}_W = \left\{ z_{nb}(t) = W_b^\top x_{nb}(t) : b \in [1, B] \right\}_{n=1}^N \tag{2}$$

with $W_b \in \mathbb{R}^{C \times Q}$ as the linearly mixing matrix of spatial filters for the band $b$, and $z_{nb}(t) \in \mathbb{R}^Q$ stands for the spatially-filtered trial with $Q$ components.

### 2.2. Time-Series Similarity through the Stein Kernel for PSD Matrices

Since each trial in the component space is band-pass filtered, the expected value of $z_{nb}(t)$ becomes zero. Hence, the band-wise covariance is computed as

$$S_{nb} = z_{nb}(t) z_{nb}(t)^\top \tag{3}$$

$$S_{nb} = W_b^\top x_{nb}(t) x_{nb}(t)^\top W_b \tag{4}$$

with $S_{nb} \in \mathbb{R}^{Q \times Q}$. The set of covariance matrices from the dataset $\mathcal{X}$ holds the linear relationships between component pairs:

$$\mathcal{S} = \left\{ S_{nb} = W_b^\top \Sigma_{nb} W_b : b \in [1, B] \right\}_{n=1}^N \tag{5}$$

being $\boldsymbol{\Sigma}_{nb} = \boldsymbol{x}_{nb}(t)\boldsymbol{x}_{nb}(t)^\top \in \mathbb{R}^{C \times C}$ the trial covariance for the $b$-th band in the input channel space. Given that each matrix in the set $\mathcal{S}$ satisfies that $\langle\, \boldsymbol{A}, \boldsymbol{S}_{nb}\boldsymbol{A}\,\rangle \leq 0$ for any $\boldsymbol{A} \neq \boldsymbol{0}$, then $\boldsymbol{S}_{nb} \in \mathcal{S}$ is positive semidefinite (PSD). As a set of PSD matrices, $\mathcal{S}$ belongs to the Riemannian manifold $\mathbb{P}_Q$, which is differentiable and a canonical higher-rank symmetric space within the real symmetric matrix space $\mathbb{S}_Q$ [21]. Therefore, there exists a PSD matrix representing each time-series in a Hilbert space of matrices endowed with a metric allowing to compare two trials.

For such a manifold, the Jensen-Bregman divergence $\delta_F(\boldsymbol{S}_{nb}, \boldsymbol{S}_{mb}) \in \mathbb{R}^+$ measures the disimilarity between two matrix elements $\boldsymbol{S}_{nb}$ and $\boldsymbol{S}_{mb}$ as:

$$\delta(\boldsymbol{S}_{nb}, \boldsymbol{S}_{mb}) = \frac{B_F(\boldsymbol{S}_{nb}, \bar{\boldsymbol{S}}_b) + B_F(\bar{\boldsymbol{S}}_b, \boldsymbol{S}_{mb})}{2} \tag{6}$$

$$B_F(\boldsymbol{S}_{nb}, \boldsymbol{S}_{mb}) = F(\boldsymbol{S}_{nb}) - F(\boldsymbol{S}_{mb}) - \langle \boldsymbol{S}_{nb} - \boldsymbol{S}_{mb}, \nabla F(\boldsymbol{S}_{mb}) \rangle_F \tag{7}$$

$$\bar{\boldsymbol{S}}_b = \frac{\boldsymbol{S}_{nb} + \boldsymbol{S}_{mb}}{2} \tag{8}$$

where $\nabla F$ denotes the gradient of the strictly convex and differentiable function $F(\cdot)$, $\langle \boldsymbol{A}, \boldsymbol{B} \rangle_F = \mathrm{tr}\left(\boldsymbol{A}^\top \boldsymbol{B}\right)$ the Frobenius inner product in the PSD matrix space, and $\mathrm{tr}\left(\cdot\right)$ the trace operator. Equation (7) defines the Bregman divergence as the positive tail of the first-order Taylor expansion of $F(\cdot)$. Then, Equation (6) corresponds to the symmetrized version of $B_F$ and holds the properties of a distance function in $\mathbb{P}_Q$ [22]. In particular, using as the convex function $F(\boldsymbol{S}_{nb}) = -\log|\boldsymbol{S}_{nb}|$ yields the Jensen-Bregman LogDet divergence defined in Equation (9), with $|\cdot|$ as the determinant operator.

$$D(\boldsymbol{S}_{nb}, \boldsymbol{S}_{mb}) = \log\left|\frac{\boldsymbol{S}_{nb} + \boldsymbol{S}_{mb}}{2}\right| - \frac{\log|\boldsymbol{S}_{nb}\boldsymbol{S}_{mb}|}{2} \tag{9}$$

Moreover, Equation (9) is invariant to affine transformations, that is, the LogDet divergence at the component space is the same in the channel space if the number of components equals the number of channels ($Q = C$). The following procedure proves the affine-invariance, assuming $\boldsymbol{W} \in \mathbb{R}^{C \times C}$ square:

$$D(\boldsymbol{S}_{nb}, \boldsymbol{S}_{mb}) = \log\left|\frac{\boldsymbol{S}_{nb} + \boldsymbol{S}_{mb}}{2}\right| - \frac{\log|\boldsymbol{S}_{nb}\boldsymbol{S}_{mb}|}{2}$$

$$D(\boldsymbol{S}_{nb}, \boldsymbol{S}_{mb}) = \log\left|\frac{\boldsymbol{W}^\top\boldsymbol{\Sigma}_{nb}\boldsymbol{W} + \boldsymbol{W}^\top\boldsymbol{\Sigma}_{mb}\boldsymbol{W}}{2}\right| - \frac{\log\left|\boldsymbol{W}^\top\boldsymbol{\Sigma}_{nb}\boldsymbol{W}\boldsymbol{W}^\top\boldsymbol{\Sigma}_{mb}\boldsymbol{W}\right|}{2}$$

$$D(\boldsymbol{S}_{nb}, \boldsymbol{S}_{mb}) = \log\left|\boldsymbol{W}^\top\frac{\boldsymbol{\Sigma}_{nb} + \boldsymbol{\Sigma}_{mb}}{2}\boldsymbol{W}\right| - \frac{\log\left|\boldsymbol{W}^\top\boldsymbol{\Sigma}_{nb}\boldsymbol{W}\right|}{2} - \frac{\log\left|\boldsymbol{W}^\top\boldsymbol{\Sigma}_{mb}\boldsymbol{W}\right|}{2}$$

$$D(\boldsymbol{S}_{nb}, \boldsymbol{S}_{mb}) = \log\left|\boldsymbol{W}^\top\right| + \log\left|\frac{\boldsymbol{\Sigma}_{nb} + \boldsymbol{\Sigma}_{mb}}{2}\right| + \log|\boldsymbol{W}|$$

$$- \frac{\log\left|\boldsymbol{W}^\top\right|}{2} - \frac{\log|\boldsymbol{\Sigma}_{nb}|}{2} - \frac{\log|\boldsymbol{W}|}{2}$$

$$- \frac{\log\left|\boldsymbol{W}^\top\right|}{2} - \frac{\log|\boldsymbol{\Sigma}_{mb}|}{2} - \frac{\log|\boldsymbol{W}|}{2}$$

$$D(\boldsymbol{S}_{nb}, \boldsymbol{S}_{mb}) = \log\left|\frac{\boldsymbol{\Sigma}_{nb} + \boldsymbol{\Sigma}_{mb}}{2}\right| - \frac{\log|\boldsymbol{\Sigma}_{nb}\boldsymbol{\Sigma}_{mb}|}{2}$$

$$D(\boldsymbol{S}_{nb}, \boldsymbol{S}_{mb}) = D(\boldsymbol{\Sigma}_{nb}, \boldsymbol{\Sigma}_{mb}) \tag{10}$$

Thanks to the distance property in Equation (6), the LogDet divergence parameterizes a radial basis function to build a similarity measure between two trials in $\mathcal{X}_W$ through their corresponding covariances in the component space, that is:

$$k(\boldsymbol{x}_{nb}(t), \boldsymbol{x}_{mb}(t)|\boldsymbol{W}_b) = e^{-\gamma_b D(\boldsymbol{S}_{nb}, \boldsymbol{S}_{mb}|\boldsymbol{W}_b)} \tag{11}$$

being $\gamma_b \in \mathbb{R}^+$ the scale parameter for the kernel from the $b$-th frequency band. Equation (11) is known as the Stein kernel for PSD matrices. Since the linear mapping $W_b$ determines the component space and the Stein kernel is affine-invariant, tuning of $W_b$ demands an optimization procedure for $Q < C$ to enhance class separability at each band.

## 2.3. Spatial Filter Optimization Using Centered Kernel Alignment

Given the kernel function in Equation (11) and the vector of target labels $y = \{y_n\}_{n=1}^N$, optimization of $W_b$ is carried out by minimizing the negative logarithm of the Centered Kernel Alignment (CKA) cost function, assessing the similarity between two random variables through the inner product of their representing kernel matrices as defined by Equation (12).

$$L(\mathcal{X}_b, y | W_b) = -\log \frac{\langle \overline{K}_b(W_b), \overline{K}_y \rangle_F}{||\overline{K}_b(W_b)||_F ||\overline{K}_y||_F} \tag{12}$$

where kernel matrices $K_b \in \mathbb{R}^{N \times N}$, with elements $k_{nm} = k(x_{nb}(t), x_{mb}(t) | W_b)$, and $K_y = \{\delta(y_n, y_m) : n, m = [1, N]\} \in \{0, 1\}^{N \times N}$ hold every pair-wise trial similarity in the component Riemmanian manifold and in the label space, respectively. $||A||_F = \sqrt{\langle A, A \rangle_F}$ corresponds to the Frobenius norm and $\delta(\cdot, \cdot)$ denotes the delta Dirac function. $\overline{K}$ stands for the centered kernel matrix asociated to $K$ and computed as in Equation (13), being $U_N$ the centering matrix, $I_N$ the $N$-order identity matrix, and $\mathbf{1}_N$ an all ones column vector. The centering operation translates all samples in the Hilbert space reproduced by the kernel $k(\cdot, \cdot)$ near the convex hull of the samples, so coping with ill-conditioning due to biased pair-wise inner products [23].

$$\overline{K} = U_N K U_N \tag{13}$$

$$U_N = I_N - \frac{1}{N} \mathbf{1}_N \mathbf{1}^\top_N \tag{14}$$

Aiming at minimizing the negative logarithm CKA between $K_b$ and the label matrix $K_y$, we consider a stochastic gradient descent approach with respect to spatial filter matrix $W_b$. Using the chain rule, the gradient of the cost function w.r.t. $W_b$ is expressed as:

$$\nabla L(W_b) = \sum_{n=1}^N \sum_{m=1}^N \frac{\partial L}{\partial k_{nm}} \frac{\partial k_{nm}}{\partial D_{nm}} \frac{\partial D_{nm}}{\partial W_b} \tag{15}$$

$$\frac{\partial L}{\partial K_b} = -\frac{\overline{K}_y}{\langle \overline{K}_b, \overline{K}_y \rangle_F} + 2 \frac{\overline{K}_b}{\langle \overline{K}_b, \overline{K}_b \rangle_F} \tag{16}$$

$$\frac{\partial k_{nm}}{\partial D_{nm}} = -\beta e^{-\gamma_b D(S_{nb}, S_{mb} | W_b)} = -\gamma_b k_{nm} \tag{17}$$

$$\frac{\partial D_{nm}}{\partial W_b} = \Sigma_{nb} W_b [2(W_b^\top (\Sigma_{nb} + \Sigma_{mb}) W_b)^{-1} - (W_b^\top \Sigma_{nb} W_b)^{-1}] \tag{18}$$

$$+ \Sigma_{mb} W_b [2(W_b^\top (\Sigma_{nb} + \Sigma_{mb}) W_b)^{-1} - (W_b^\top \Sigma_{mb} W_b)^{-1}]$$

Therefore, maximizing the alignment between samples and their label kernels yields a matrix $W_b$ rotating the channel data so that the Riemmanian space of component covariances better discriminates the given classes in the frequency band $b$.

## 2.4. Assembling of Multiple Kernel Representations

Given the kernel function in Equation (11) providing a metric for a frequency band, the knowledge from each band can be merged into a single metric using a multiple kernel learning (MKL) approach.

For the set of matrices $\{\boldsymbol{K}_b \in \mathbb{R}^{N \times N}\}_{b=1}^B$, the linear combination in Equation (19) corresponds to a weighted concatenation of the Hilbert space features reproduced by each kernel [24].

$$\boldsymbol{K}_\mu = \sum_{b=1}^B \mu_b \boldsymbol{K}_b \tag{19}$$

$$k_{\boldsymbol{\mu}}(\boldsymbol{x}_n(t), \boldsymbol{x}_m(t)) = \sum_{b=1}^B \mu_b k(\boldsymbol{x}_{nb}(t), \boldsymbol{x}_{mb}(t)|\boldsymbol{W}_b) \tag{20}$$

Aiming to preserve the positive definiteness of a kernel function, the vector of weights $\boldsymbol{\mu} = \{\mu_b\}_{b=1}^B$ must belong to subset of positive one-valued norm vectors, that is, $\boldsymbol{\mu} \in \{\boldsymbol{\mu} \in \mathbb{R}^B : ||\boldsymbol{\mu}||_2 = 1, \boldsymbol{\mu} \geq \boldsymbol{0}\}$. Finding the optimal $\boldsymbol{\mu}$ is posed as the maximization of the CKA cost between the kernel matrices $\boldsymbol{K}_\mu = \{k_{\boldsymbol{\mu}}(\boldsymbol{x}_n(t), \boldsymbol{x}_m(t))\}_{nm=1}^N$ and $\boldsymbol{K}_y$ while constraining the weight values as follows:

$$\max_{\boldsymbol{\mu} \in \mathbb{R}^B} \frac{\langle \overline{\boldsymbol{K}}_\mu, \overline{\boldsymbol{K}}_y \rangle_F}{||\overline{\boldsymbol{K}}_\mu||_F ||\overline{\boldsymbol{K}}_y||_F} \tag{21}$$
$$s.t. ||\boldsymbol{\mu}||_2 = 1$$
$$\boldsymbol{\mu} \geq \boldsymbol{0}$$

The optimization problem in Equation (21) is rewritten as a straightforward quadratic optimization problem with linear constraints [24]:

$$\min_{\boldsymbol{v} \in \mathbb{R}^B} \boldsymbol{v}^\top \boldsymbol{M} \boldsymbol{v} - 2\boldsymbol{v}^\top \boldsymbol{a} \tag{22}$$
$$s.t. \boldsymbol{v} \geq \boldsymbol{0}$$

where vector $\boldsymbol{a} = \left\{\langle \overline{\boldsymbol{K}}_b, \overline{\boldsymbol{K}}_y \rangle_F\right\}_{b=1}^B \in \mathbb{R}^B$ and matrix $\boldsymbol{M} = \left\{\langle \overline{\boldsymbol{K}}_b, \overline{\boldsymbol{K}}_{b'} \rangle_F\right\}_{bb'=1}^B \in \mathbb{R}^{B \times B}$ account for the alignment with the supervised information and between the input kernels, respectively. Lastly, the solution weighting vector is achieved by normalizing $\boldsymbol{v}$ as:

$$\boldsymbol{\mu} = \frac{\boldsymbol{v}}{||\boldsymbol{v}||_2} \tag{23}$$

Therefore, the weights given by Equation (23) rank each frequency band according its similarity with the provided labels, so that the smaller the value, the leaser the contribution to build the kernel in Equation (19). Further, the introduced MKL gathers the band-wise covariance matrices from different Riemannian manifolds into a single supervised reproduced kernel Hilbert space favoring separability of mental states.

## 3. Experimental Setup

### 3.1. Dataset IIa from BCI Competition IV (BCICIV2a)

To test our proposal, we use the dataset IIa of the BCI Competition IV, provided by the BCI Laboratory at the Graz University of Technology (http://www.bbci.de/competition/iv/). The dataset BCICIV2a comprises EEG trials from nine subjects while executing four motor imagery tasks, namely, left hand, right hand, foot, and tongue. The participants imagined each movement 72 times following a visual cue, resulting in $N = 288$ trials. For each trial, twenty-two EEG electrodes ($C = 22$) distributed over the scalp (as seen in Figure 1a) recorded the brain activity at a sampling frequency of 250 Hz during six seconds. The trial started with an auditory signal, followed by a black fixation cross, that warned the subject for the upcoming cue. An arrow pointing towards the movement during 1.25 s indicated the task to perform. The trial ended four seconds after the fixation cross disappeared as Figure 1b shows. A break period of about 1.5 s allows the participant to rest before starting the next

trial. Taking into account the cue onset and the short-lasting imagined movement, the methodology classifies each recording only using the period between 2.5 and 4.5 s, that is, $T = 500$ samples.

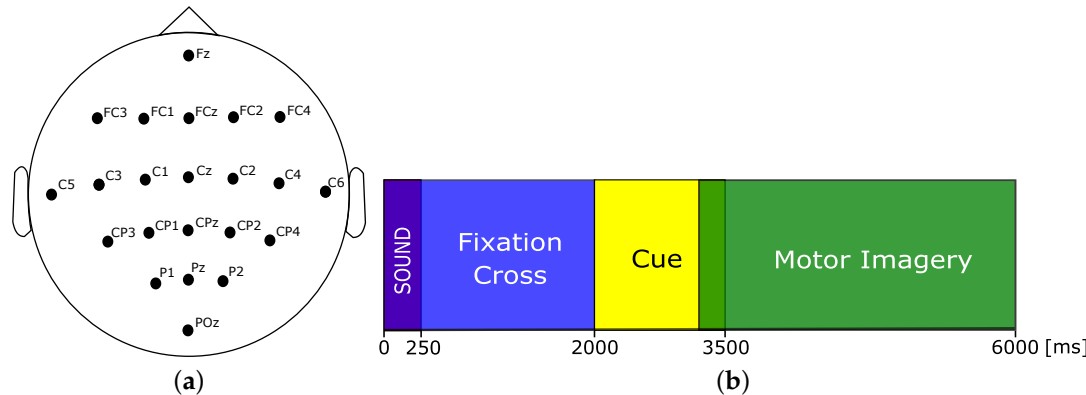

**Figure 1.** BCI competition IV acquisition setup. (**a**) EEG montage. (**b**) Paradigm time scheme.

### 3.2. Proposed BCI Methodology

Figure 2 illustrates the five stages of the proposed EEG classification methodology for BCI. Firstly, $B = 17$ band-pass filters decompose each trial using five-ordered Butterworth filters of 4 Hz bandwidth and overlapping 2 Hz within [4,40] Hz [25]. Secondly, the spatial filters, optimized in Section 2.3, project the band-wise covariance matrix into the lower-dimensional Riemannian manifold $\mathbb{P}_Q$. Thirdly, the Stein kernel assesses the similarity between the projected test and training covariances, relying on Equation (11). Further, the multikernel linearly combines the 17 similarities for a test trial into a single kernel value. Lastly, a support vector machine, fed by the learned kernel, labels the recordings into one of the four classes according to a one-vs-one scheme.

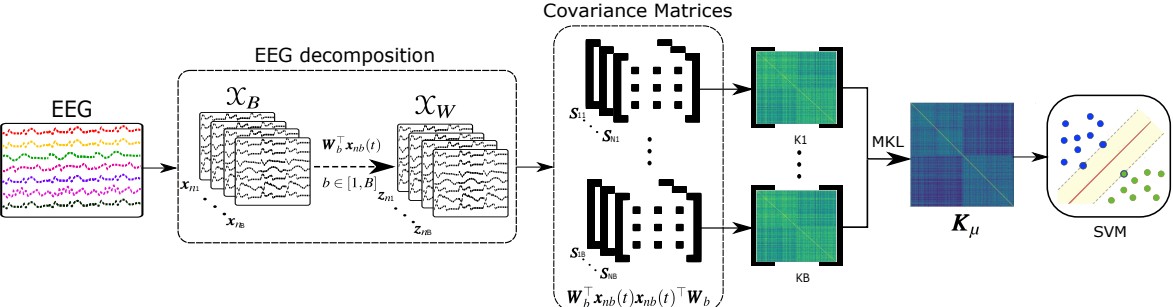

**Figure 2.** Proposed EEG classification methodology based on filter-banked Stein-kernel trial similarities.

The proposed methodology contains three sets of parameters, namely, the band-wise spatial filters, the kernel bandwidth, and the kernel mixing weights. For the former two, the gradient descent algorithm, updating the parameter according to Equation (18), finds the channel-to-component space projections and gamma values enhancing class separability at each band. For the latter, the solution to the quadratic programming problem in Equation (22) defines the contribution of each spectral component to the supervised discrimination task. Besides, the single hyperparameter of the proposal corresponds to $Q$, defining the number of columns for matrices $W_b$, that is, the number of spatial filters. A cross-validated grid-search approach fixes the optimal $Q$ by maximizing the classification performance within the range $[2, 10]$. Specifically, we assess the performance through Cohen's kappa score defined in Equation (24), where $p_o$ corresponds to the empirical probability of agreement on the label assigned to any sample (the observed agreement ratio), and $p_e$ determines the expected agreement at classification by chance.

$$\kappa = \frac{p_o - p_e}{1 - p_e} \qquad (24)$$

## 4. Results

### 4.1. Performance Results

Figure 3 presents the subject-wise grid search results for tuning the number of spatial filters. Tuning curves evidence that performance decreases for either a very low or very high number of components. In the first case, the resulting model misses class separability patterns, due to the low flexibility in the lowest dimensional manifolds. In the second case, the methodology overfits the training data, since the component space becomes sparse, then drops the validation performance. Since four to eight components improve the classification performance in comparison to the input channel space, the proposed methodology reduces Riemmannian manifold dimension while enhancing MI discrimination.

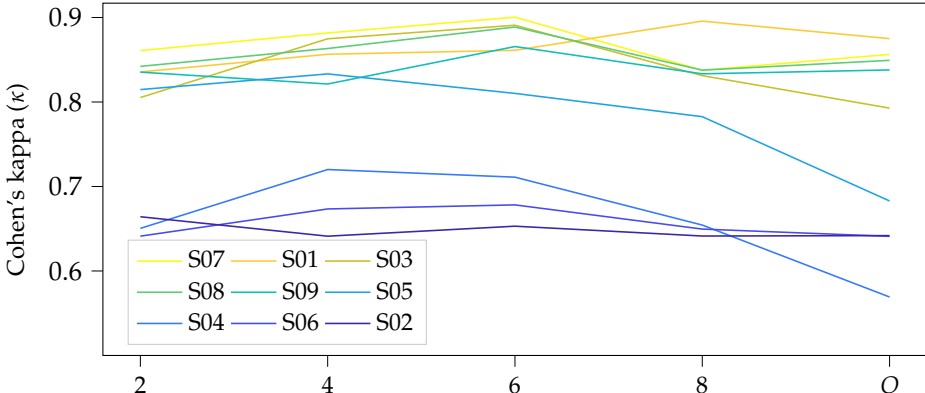

**Figure 3.** Cross-validated classification performance along the number of components.

Table 1 presents the mean kappa value using different methods tested in the BCICIV2a dataset, with the highest kappa value in bold for each subject. The last column holds the *p*-value for a paired *t*-test between the proposed and each compared approaches. Subjects are sorted according the score attained by the challenge winner. Note that MKSSP outperforms other approaches at the worst performing subjects (namely S06, S05, S02, and S01). Specifically, the proposal favors the most the subject S05, as the kappa score raises between 5% and 57%. Despite MKSSP performs at subjects S09, S03, S08, and S07, the best approach on such subjects are considerably biased towards them. Furthermore, MKSSP reaches the highest average kappa score of 0.82, with significant differences in most of the comparisons. Hence, the proposed methodology collects and combines discriminative features from different spaces in a single representation, enhancing the separability of different mental tasks.

**Table 1.** Mean kappa scores attained by compared approaches at each subject from the BCICIV2a dataset. The last two columns present the average kappa and the *t*-test *p*-value between Multi-Kernel Stein Spatial Patterns (MKSSP) and the corresponding approach. In bold highest kappa values, in italic *p*-values < 5%.

| Approach | S06 | S05 | S02 | S04 | S01 | S09 | S03 | S08 | S07 | $\kappa$ | *p*-Value |
|---|---|---|---|---|---|---|---|---|---|---|---|
| Challenge winner [26] | 0.27 | 0.40 | 0.42 | 0.48 | 0.68 | 0.61 | 0.75 | 0.75 | 0.77 | $0.57 \pm 0.17$ | *0.0002* |
| SUSS-SRKDA [27] | 0.35 | 0.56 | 0.51 | 0.68 | 0.83 | 0.75 | 0.88 | 0.84 | 0.90 | $0.70 \pm 0.18$ | *0.0179* |

**Table 1.** *Cont.*

| Approach | S06 | S05 | S02 | S04 | S01 | S09 | S03 | S08 | S07 | $\kappa$ | *p*-Value |
|---|---|---|---|---|---|---|---|---|---|---|---|
| CBN [28] | 0.42 | 0.78 | 0.51 | **0.85** | 0.69 | 0.45 | 0.87 | **0.97** | 0.54 | $0.68 \pm 0.19$ | 0.0577 |
| KPCA with CILK [29] | 0.37 | 0.26 | 0.46 | 0.44 | 0.71 | 0.61 | 0.76 | 0.75 | 0.79 | $0.57 \pm 0.18$ | *0.0009* |
| PSO [30] | 0.53 | 0.62 | 0.62 | 0.77 | 0.87 | 0.76 | **0.90** | 0.82 | 0.80 | $0.74 \pm 0.12$ | *0.0282* |
| CSP-FLS [31] | 0.37 | 0.35 | 0.54 | 0.52 | 0.74 | 0.80 | **0.90** | 0.86 | 0.82 | $0.66 \pm 0.20$ | *0.0146* |
| EMD+Riemann [32] | 0.34 | 0.36 | 0.24 | 0.68 | 0.86 | 0.82 | 0.70 | 0.75 | 0.66 | $0.60 \pm 0.21$ | *0.0050* |
| CSP/AM-BA-SVM [33] | 0.41 | 0.58 | 0.55 | 0.60 | 0.87 | 0.80 | 0.89 | 0.84 | 0.88 | $0.71 \pm 0.17$ | *0.0147* |
| Dempster–Shafer [34] | 0.57 | 0.67 | 0.59 | 0.72 | 0.78 | 0.88 | 0.85 | 0.86 | 0.81 | $0.75 \pm 0.11$ | *0.0084* |
| Functional brain [35] | 0.61 | 0.63 | 0.54 | 0.70 | 0.77 | 0.86 | 0.84 | 0.84 | 0.77 | $0.73 \pm 0.11$ | *0.0036* |
| CNN-LSTM [36] | 0.66 | 0.77 | 0.54 | 0.78 | 0.85 | **0.90** | 0.87 | 0.83 | **0.95** | $0.80 \pm 0.12$ | 0.3313 |
| sDPLM [15] | 0.36 | 0.34 | 0.49 | 0.49 | 0.75 | 0.76 | 0.76 | 0.76 | 0.68 | $0.60 \pm 0.17$ | *0.0008* |
| uDPLM [15] | 0.36 | 0.30 | 0.49 | 0.47 | 0.76 | 0.76 | 0.76 | 0.76 | 0.69 | $0.59 \pm 0.18$ | *0.0012* |
| **Proposed MKSSP** | **0.68** | **0.83** | **0.66** | 0.72 | **0.90** | 0.87 | 0.89 | 0.89 | 0.90 | $\mathbf{0.82 \pm 0.09}$ | – |

To evaluate the proposal sensitivity to noise, we performed a nested cross-validation test by adding several noise levels to the EEG signals. For each subject, 70% of the trials were considered to learn the model parameters at the optimal number of components using a five fold inner cross-validation. The remaining 30% was firstly contaminated with additive Gaussian noise from 0 dB to 40 dB Signal-to-Noise Ratios (SNR). Then, the class-wise performance was computed from the contaminated trials while varying the SNR. Figure 4 illustrates the resulting class-wise true positive ratio (TPR) with the subjects ordered from best to worst performance, as in Figure 3.

Overall, the proposed MKSSP reaches a stable performance from 30 dB. Within 20 to 30 dB, most of the subjects and classes increase their TPR to reach the optimal behavior. However, the overall performance drops for SNR worse than 20 dB. At the subject level, the best performing ones yield the same behavior for all classes, that is, a low TPR for SNR < 20 dB and the optimal TPR for SNR > 30 dB. On the contrary, the worse performing subjects exhibit a TPR biased towards Foot and Tongue when the SNR < 20 dB. For SNR > 30 dB, the hand-related classes evidenced higher TPR values for the best performing subjects than for the worse ones. Such findings suggest that the additive noise mostly hinders the discriminative patterns related to Left and Right. In turn, the Foot class presents a similar performance for SNR > 30 dB among all the subjects, implying patterns that are harder to reveal. Therefore, the proposed classification methodology reliably operates at SNR levels better than 30 dB with less biased results.

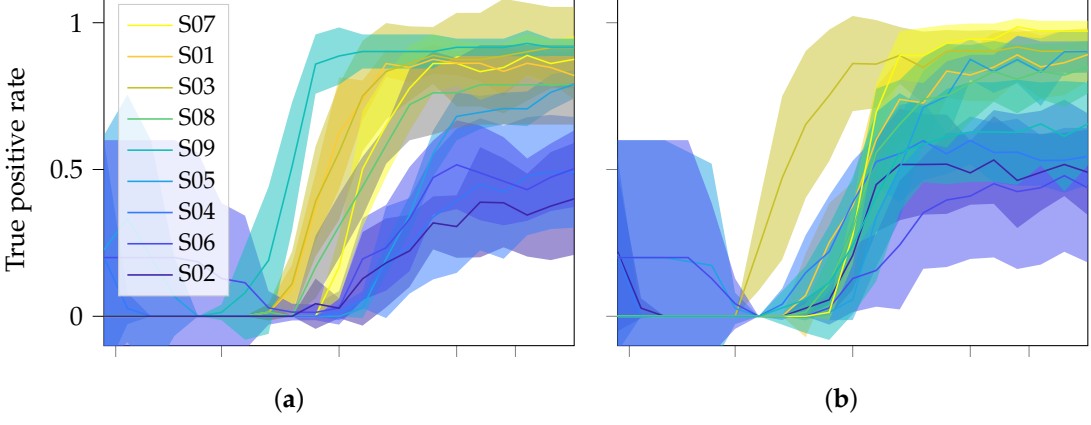

**Figure 4.** *Cont.*

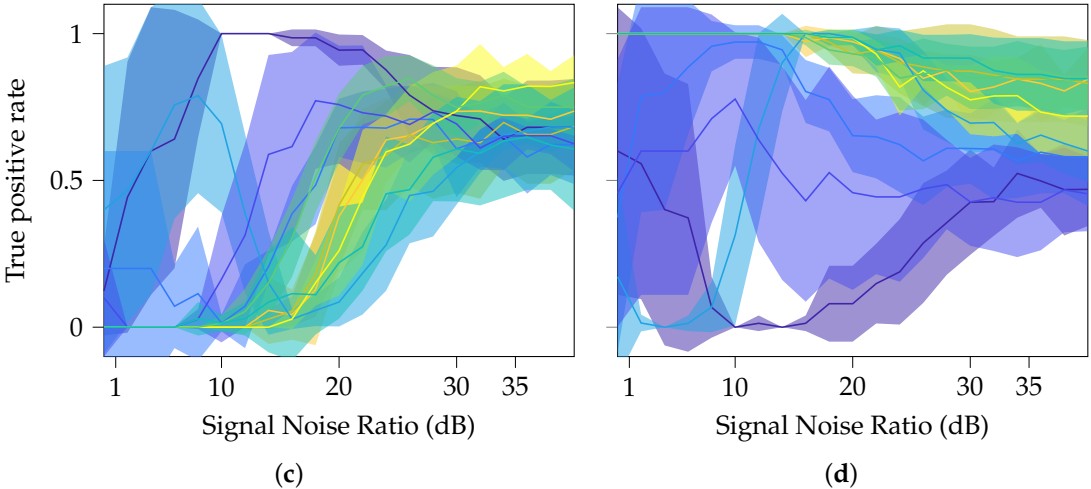

**Figure 4.** Noise sensitivity test per class for additive Gaussian noise using the optimal parameter set. (**a**) Left class. (**b**) Right class. (**c**) Foot class. (**d**) Tongue class.

## 4.2. Model Interpretability

We evaluate the model interpretability from three points of view: the data, spectral, and spatial representations. For analyzing the data representation, Figures 5 and 6 illustrate the three-dimensional mapping of $K_\mu$ using Kernel Principal Component Analysis (KPCA) for the best- (S07) and worst-performing (S02) subjects, respectively. At the first two KPCA components of S07, MKSSP discriminated Left and Right from Foot and Tongue classes. Those two components also allow classifying between Left and Right, whereas the third one separated Foot and Tongue. Despite being less evident, MKSSP identified three trial groups from Subject S02, namely Foot, Tongue, and hands. The first two KPCA components better separated Foot from the remaining classes, while the first and third ones enhanced Tongue discrimination.

As Table 1 presented, subject S05 benefited the most from the proposed MKSSP, in comparison with the challenge winner. For such a subject, Figure 7 compares the trial data distribution resulting from four approaches: CSP, with a single 8-to-30 Hz band-pass filter, and Filter-Banked CSP (FBCSP), with filters as in Section 3.2, representing the baseline spatial-filtering and spectro-spatial-filtering techniques. MKSSP without spatial filtering (i.e., $W$ is the $C \times C$ identity matrix $I_C$) and with the optimal number of components $Q^* = 4$ contrast the former approaches as spectral and spectro-spatial representations. For the sake of visualization, PCA and KPCA map trial features to two dimensions for CSP-bases and MKSSP-based representations, respectively. It is worth noting that CSP features without spectral filtering (top-left) and MKSSP without spatial filtering (bottom left) lacked any class separability. In turn, FBCSP separated Left and Right but mixed Foot and Tongue within the four classes. Lastly, optimal MKSSP not only increased Left and Right distance but also unraveled Foot and Tongue.

Regarding the spectral representation, Figure 8 illustrates the MKL weights of each band per subject, descending-ordered on the Y-axis (from best to worst based on the kappa score). A visual inspection of spectral weights segment subjects into two groups: Subjects S07, S01, S03, S08, and S09; and subjects S05, S04, S06, and S02. The first group, holding the best performing subjects, concentrates weights at frequencies lower than 26Hz. Besides, the reduced number of highlighted spectral bands translates into a computational burden reduction. The second group, with kappa lower than 0.83, widely spreads all over the signal bandwidth, without accentuating any spectral filter. Accordingly, the weight distribution marks the MKSSP performance on the subject.

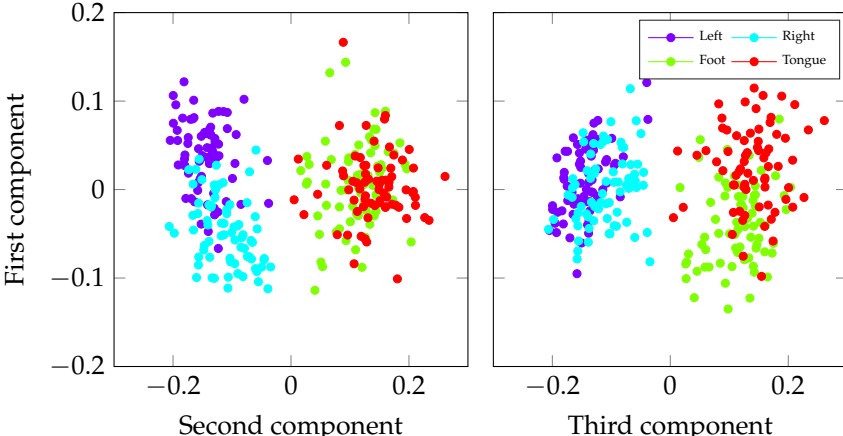

**Figure 5.** Resulting MKSSP kernel projected into three Kernel Principal Component Analysis (KPCA) components for the best performing subject (S07).

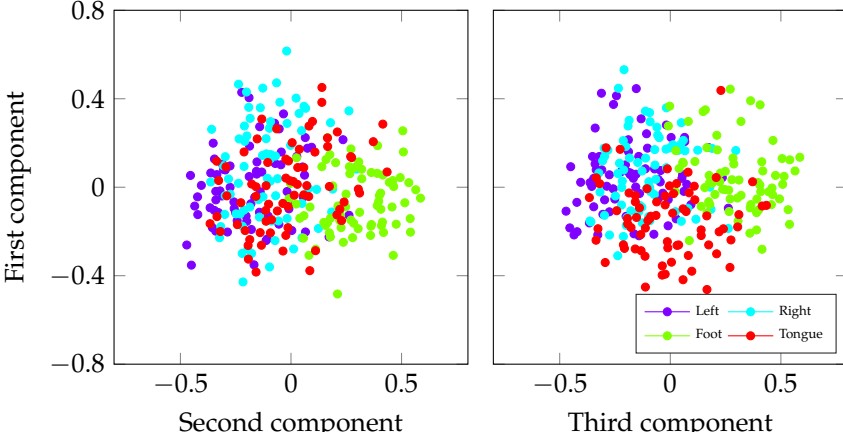

**Figure 6.** Resulting MKSSP kernel projected into three KPCA components for the worst-performing subject (S02).

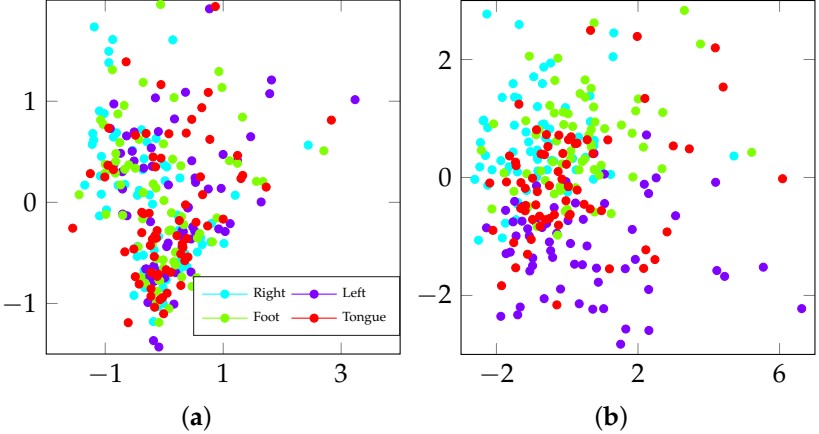

(**a**)          (**b**)

**Figure 7.** *Cont.*

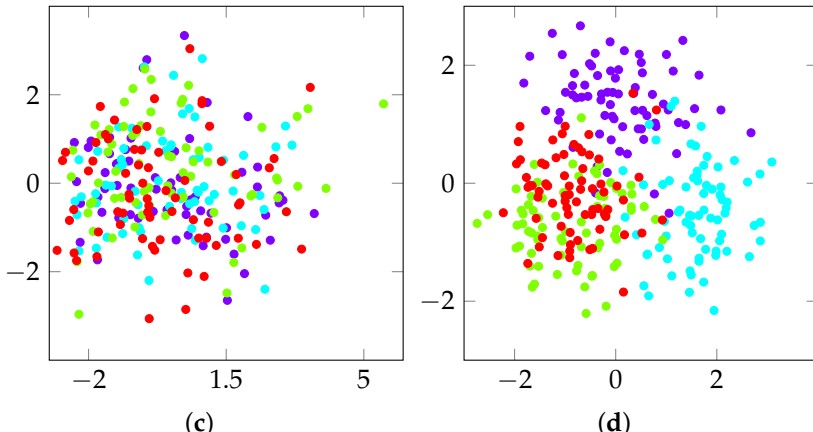

**Figure 7.** Projections of subject S05 trials using four spatial patterns approaches. Either PCA or KPCA maps to a 2D space features from (FB) CSP or MKSSP, respectively. (**a**) CSP. (**b**) FBCSP. (**c**) MKSSP ($W = I_C$). (**d**) MKSSP ($Q^* = 4$).

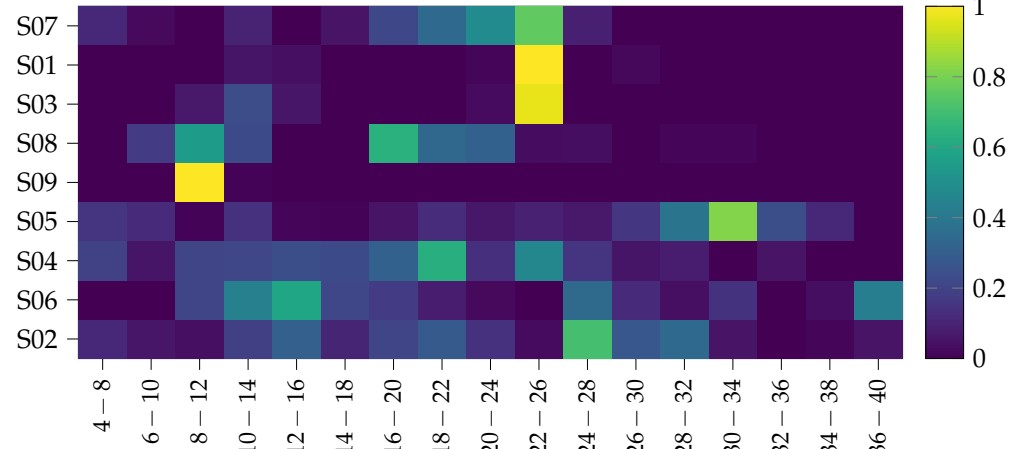

**Figure 8.** Multiple kernel learning (MKL) kernel weights per frequency band. Subjects are sorted according to the kappa score from the best (top) to the worst (bottom) in Y-axis. X-axis indexes each band-pass filter in the bank.

For the spatial representation, Figure 9 depicts the spatial patterns of the best and worst performing subjects for MKSSP (S07 and S02, respectively) and the most favored one (S05), since such topographic maps represent the projection of estimated brain activity sources to electrodes [7]. For each subject, the top and bottom row hold the least and most weighted frequency band, respectively, while columns sort the first four components. For subject S07, the first two columns highlight a contralateral activity localized over the motor cortex region at the most relevant frequency band, which may be associated with the Right and Left classes [37]. On the contrary, the subject S02 lacks any activity over such cortex, yielding patterns without MI interpretation. In addition, the bottom frequency band of subject S05 relates the more to imagined movements than the top one, since the first two components better localize brain activity over hand movement regions. Lastly, the activity concentrated over the center vertex stands out in the third component and bottom band of all the subjects and explains Foot movement imagination [37].

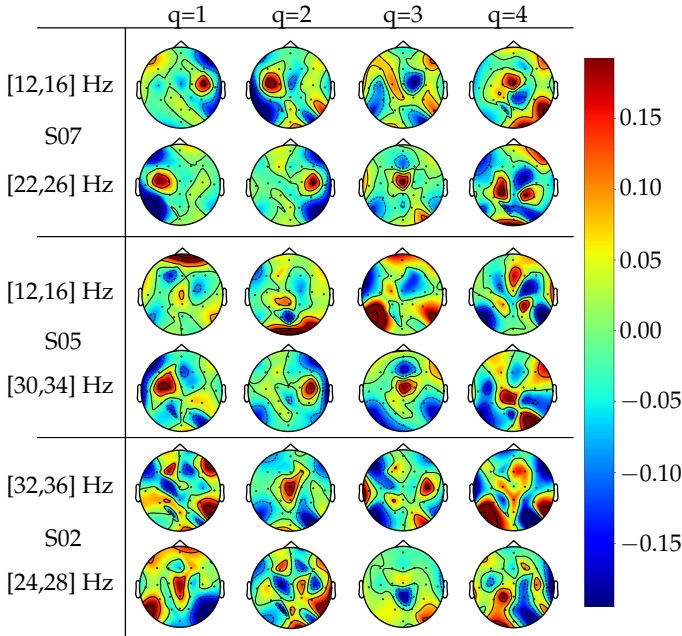

**Figure 9.** First four spatial patterns of the best performing (S07), worst performing (S02), and most improved (S05) subjects computed from MKSSP. For each subject, the top and bottom row hold the least and most weighted frequency band, respectively.

## 5. Discussion and Concluding Remarks

This work presents a methodology for the classification of SPD matrices, termed MKSSP and outlined in Figure 2. MKSSP performs a supervised nonlinear dimension reduction with the following benefits for the discrimination of multichannel EEG trials: firstly, the projection of PSD matrices into a smaller-dimensional Riemannian manifold allows preserving the geometry of the channel space. Secondly, the introduced CKA-based supervised learning optimizes the linear mapping between Riemannian manifolds to exploit the nonlinear relationships that discriminate classes. In this regard, we benefit from the Stein kernel parameterized by the Jensen-Bregman LogDet divergence, which is devoted to comparing SPD matrices. Thirdly, the spectral decomposition highlights discriminative patterns hidden by large-power frequencies. Since each band yields a single kernel, the MKL scheme gathering several kernel representations augmented the class separability.

Regarding the performance evaluation, we considered the four-class EEG signals from the Dataset IIa of the BCI Competition IV. According to Figure 3, illustrating the kappa score across the number of components, the dimension reduction carried out by MKSSP outperforms the kappa score of the input channel-space at all subjects, so avoiding the curse of dimensionality. Moreover, Table 1 evidences a significant improvement in class discrimination in comparison with state-of-the-art approaches. Specifically, MKSSP considerably increases the kappa score of usually low-performing subjects, such as S05, indicating that nonlinear relationships between channels hold discriminant MI information. While spatial filtering approaches, that map into euclidean spaces, hinder the nonlinearity, MKSSP takes decode it into a single kernel to enhance class separability.

Concerning the noise sensitivity, the classification results in Figure 4 guarantee reliable performances for SNR levels better than 30 dB. SNR level within 20 to 30 dB compromises the class-wise TPR. In fact, the proposed MKSSP reaches the best TPR on Right and Left-hand classes, possibly due to their contralateral nature. Despite the performance of around 80% TPR, the identification of the Feet movement remains challenging compared to other movements.

For the model interpretability, Figures 5 and 6 illustrate the kernel resulting from Equation (19) in a 2D KPCA-based projection for subjects S07 and S02 (the best and worst-performing, respectively). Note that MKSSP finds a space suitably separating classes for the subject S07. Despite the lowest performance of subject S02 and mixed Left and Right trials, the proposed methodology enhances

the discrimination of three groups, namely, Foot, Tongue, and both hands. Such a result proves the capability of MKSSP to distinguish mental tasks despite the spatially close activity sources. Furthermore, Figure 7, comparing four feature sets for subject S05, indicates that CSP-based approaches lack separability as they miss geometric nonlinear channel relationships. Although including spectral decomposition, MKSSP without dimension reduction tangles the trials, due to the high dimension in the covariance matrices hinders discriminative patterns. On the contrary, MKSSP with the optimal number of components enhances class separation, agreeing with the test classification results, because the manifold information benefits the MI task identification.

In the spectral relevance, Figure 8 depicts the band-wise MKL weights per subject. The weight distribution highlights the between-subject variability in terms of spectral relevance. For the best performing subjects, MKSSP concentrates the kernel weights in fewer bands than for the worst ones. Therefore, the proposed methodology avoids the computation of unnecessary frequency components for discriminating MI tasks. Further, the weight distribution may predict the subject BCI performance.

Regarding the spatial representation, Figure 9 presents the resulting Stein spatial patterns for subjects S07, S05, and S02. Only the patterns corresponding to the least (top) and most (bottom) weighted bands are displayed. On the one hand, the least weighted bands yield wider activity sources without neurophysiological interpretability. On the other hand, MKSSP mostly highlights bands with spatial patterns physiologically related to MI tasks. For instance, the contralateral activities, typical of left and right-hand movement, emerge from the first two patterns of subjects S07 and S05. In addition, the third pattern for all subjects finds activity close to the central vertex related to the foot movement. Consequently, MKSSP correctly decodes spatially interpretable patterns while spectrally highlighting the bands containing them.

Future work considers two research directions. Firstly, we will work on the development of a BCI ability estimator based on the MKSSP methodology to evaluate the subject efficiency, since up to 30% of people underdevelop the coordination skills after training, hampering the spreading of this kind of system [38]. Secondly, we will consider a transfer learning scheme for training the MKSSP-based representation to achieve subject-independent BCI systems with reduced training times.

**Author Contributions:** Conceptualization, D.C.-P. and S.G.-N.; methodology, D.C.-P.; software, S.G.-N.; validation, D.C.-P., Á.O.-G. and S.G.-N.; formal analysis, D.C.-P. and S.G.-N.; investigation, S.G.-N.; resources, Á.O.-G.; data curation, D.C.-P.; writing—original draft preparation, S.G.-N.; writing—review and editing, D.C.-P. and Á.O.-G.; supervision, D.C.-P. and Á.O.-G.; project administration, Á.O.-G.; funding acquisition, Á.O.-G. All authors have read and agreed to the published version of the manuscript.

**Funding:** This research was funded by the project with code 111080763051 by MinCiencias and Universidad Tecnológica de Pereira and by the Vice-rectory for research from Universidad Tecnológica de Pereira by the project with code E6-20-3.

**Acknowledgments:** This work was supported by the research project "Herramienta de apoyo al diagnóstico del TDAH en niños a partir de múltiples características de actividad eléctrica cerebral desde registros EEG" with code number 111080763051 and funded by MinCiencias. Authors also thank to the Vice-rectory for research from Universidad Tecnológica de Pereira for supporting the development of this work.

**Conflicts of Interest:** The authors declare no conflict of interest.

## Abbreviations

The following abbreviations are used in this manuscript:

| | |
|---|---|
| BCI | Brain–Computer Interface |
| EEG | Electroencephalography |
| MI | Motor Imagery |
| CSP | Common Spatial Pattern |
| SPD | Symmetric Positive Definite |
| CKA | Centered Kernel Alignment |
| MKL | Multiple Kernel Learning |
| MKSSP | Multi-Kernel Stein Spatial Patterns |

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
