# Peer review of "Multiple Kernel Stein Spatial Patterns for the Multiclass Discrimination of Motor Imagery Tasks"

_applsci, doi:10.3390/app10238628_

Round 1
Reviewer 1 Report
The paper entitled “Multiple kernel Stein spatial patterns for the multiclass discrimination of motor imagery tasks” proposes a method to classify the EEG signal for the brain-compter interface. They first decompose the signal into spectrally independent signals. Then each signal is mapped into low-dimensional covariance matrices. Then the similarities from spectral decomposition are used as input for the classifier. The paper is well written and technically sounds. There are a few questions that would be good to be answered.
- Please state clearly the novelty of this work compared to the previous works.
- It would be good to investigate the sensitivity of the method at different noise levels.
Author Response
We want to thank you for the comments over our manuscript. All your suggestions have been very helpful. We made a large effort to follow them thoughtfully to improve the document quality.
- Please state clearly the novelty of this work compared to the previous works.
Answer Thanks for the appropriate advice. In order to make clearer the proposal novelty compared to previous approaches, we have re-stated the end of the Introduction section as follows:
For dealing with the above issues, this work proposes a methodology for the classification of SPD matrices with three main contributions: Firstly, we introduce a dimension reduction for SPD matrices. Contrarily to other approaches, the approach uses the provided labels to learn the mapping that not only preserves the data structure but also favors the class discrimination thanks to a supervised learning scheme. Secondly, we propose two kernel mappings to decode the non-linear relationships that enhance the separability of classes at the component level, namely, the Stein kernel devoted to SPD matrices, and the linear combination of kernel functions unifying the knowledge from different spectral bands.
The methodology, termed Multi-Kernel Stein Spatial Patterns (MKSSP), includes a spectral decomposition, spatial filtering, and supervised kernel learning stages to enhance the discrimination of multichannel EEG signals. Firstly, a bank of bandpass filters decomposes the multichannel signals, from which MKSSP computes the covariance matrices. Then, the spatial filtering projects the band-wise covariances into smaller-dimensional Riemannian manifolds by maximizing a supervised kernel-based cost function. The Jensen-Bregman LogDet Divergence, devoted to SPD matrices, parameterizes the considered kernel pair-wise comparing EEG trial at each band. Finally, a multiple-kernel yields a single similarity measure that feeds a classifier. The centered kernel alignment (CKA) works as the cost function for training the dimension reduction and the multi-kernel. We tested our methodology in the Dataset IIa of the BCI Competition IV, outperforming the state-of-the-art in the four task MI classification for most of the subjects and achieving significant improvements for the ones with usual low-performance scores.
Aiming to gain insight into the methodology benefits, we analyze the methodology from four perspectives. The data analysis evidences the class separation. The noise sensitivity validates the robustness and class unbiasing. The spectral and spatial points of view explain the subject variability and model interpretability, respectively.
- ``It would be good to investigate the sensitivity of the method at different noise levels.''\\
Answer In order to follow the advice, we carried out a class-wise noise sensitivity analysis, which we found interesting for multiclass problems. Figure 4 presents the attained true positive rates while varying the signal to noise ratio of an additive Gaussian noise. Besides, the subsection 4.1 now describes and comments the most relevant findings as follows:
To evaluate the proposal sensitivity to noise, we performed a nested cross-validation test by adding several noise levels to the EEG signals. For each subject, 70\% of the trials were considered to learn the model parameters at the optimal number of components using a five fold inner cross-validation. The remaining 30\% was firstly contaminated with additive Gaussian noise from 0$dB$ to 40$dB$ Signal-to-Noise Ratios (SNR). Then, the class-wise performance was computed from the contaminated trials while varying the SNR. Figure 4 illustrates the resulting class-wise true positive ratio (TPR) with the subjects ordered from best to worst performance, as in Figure 3.
Overall, the proposed MKSSP reaches a stable performance from 30 $dB$. Within 20 to 30 $dB$, most of the subjects and classes increase their TPR to reach the optimal behavior. However, the overall performance drops for SNR worse than 20 $dB$. At the subject level, the best performing ones yield the same behavior for all classes, that is, a low TPR for SNR $<$ 20 $dB$ and the optimal TPR for SNR $>$ 30 $dB$. On the contrary, the worse performing subjects exhibit a TPR biased towards Foot and Tongue when the SNR $<$ 20 $dB$. For SNR $>$ 30 $dB$, the hand-related classes evidenced higher TPR values for the best performing subjects than for the worse ones. Such findings suggest that the additive noise mostly hinders the discriminative patterns related to Left and Right. In turn, the Foot class presents a similar performance for SNR $>$ 30 $dB$ among all the subjects, implying patterns that are harder to reveal. Therefore, the proposed classification methodology reliably operates at SNR levels better than 30 $dB$ with less biased results.
Also, the section 5 discusses the principal aspects of the noise analysis as:
Concerning the noise sensitivity, the classification results in Figure 4 guarantee reliable performances for SNR levels better than 30 $dB$. SNR level within 20 to 30 $dB$ compromises the class-wise TPR. In fact, the proposed MKSSP reaches the best TPR on Right and Left-hand classes, possibly due to their contralateral nature. Despite the performance of around 80\% TPR, the identification of the Feet movement remains challenging compared to other movements.
Reviewer 2 Report
The paper deals with classifiying noisy EEG signals, is written in a readable way, comprehensible.
It has proposed a valid method that is quite thoroughly experimentaly justified.
I would definitely recommend this for publication, but I recommend to improve introduction and conlusion - to include better explanation of the contributions of this work and emphasize what is new a different from existing approaches.
Author Response
We want to thank you for the comments over our manuscript. All your suggestions have been very helpful. We made a large effort to follow them thoughtfully to improve the document quality.
- ``I would definitely recommend this for publication, but I recommend to improve introduction and conlusion - to include better explanation of the contributions of this work and emphasize what is new a different from existing approaches.''
Answer Thanks for the appropriate advice. In order to make clearer the novelty and contributions, we have re-stated the end of the Introduction section as follows:
For dealing with the above issues, this work proposes a methodology for the classification of SPD matrices with three main contributions: Firstly, we introduce a dimension reduction for SPD matrices. Contrarily to other approaches, the approach uses the provided labels to learn the mapping that not only preserves the data structure but also favors the class discrimination thanks to a supervised learning scheme. Secondly, we propose two kernel mappings to decode the non-linear relationships that enhance the separability of classes at the component level, namely, the Stein kernel devoted to SPD matrices, and the linear combination of kernel functions unifying the knowledge from different spectral bands.
The methodology, termed Multi-Kernel Stein Spatial Patterns (MKSSP), includes a spectral decomposition, spatial filtering, and supervised kernel learning stages to enhance the discrimination of multichannel EEG signals. Firstly, a bank of bandpass filters decomposes the multichannel signals, from which MKSSP computes the covariance matrices. Then, the spatial filtering projects the band-wise covariances into smaller-dimensional Riemannian manifolds by maximizing a supervised kernel-based cost function. The Jensen-Bregman LogDet Divergence, devoted to SPD matrices, parameterizes the considered kernel pair-wise comparing EEG trial at each band. Finally, a multiple-kernel yields a single similarity measure that feeds a classifier. The centered kernel alignment (CKA) works as the cost function for training the dimension reduction and the multi-kernel. We tested our methodology in the Dataset IIa of the BCI Competition IV, outperforming the state-of-the-art in the four task MI classification for most of the subjects and achieving significant improvements for the ones with usual low-performance scores.
Aiming to gain insight into the methodology benefits, we analyze the methodology from four perspectives. The data analysis evidences the class separation. The noise sensitivity validates the robustness and class unbiasing. The spectral and spatial points of view explain the subject variability and model interpretability, respectively.
Further, we modified the first part of the discussion section so that the contribution of each proposed processing stage in the methodology becomes evident:
This work presents a methodology for the classification of SPD matrices, termed MKSSP and outlined in Figure 2. MKSSP performs a supervised non-linear dimension reduction with the following benefits for the discrimination of multichannel EEG trials: Firstly, the projection of PSD matrices into a smaller-dimensional Riemannian manifold allows preserving the geometry of the channel space. Secondly, the introduced CKA-based supervised learning optimizes the linear mapping between Riemannian manifolds to exploit the non-linear relationships that discriminate classes. In this regard, we benefit from the Stein kernel parameterized by the Jensen-Bregman LogDet divergence, which is devoted to comparing SPD matrices. Thirdly, the spectral decomposition highlights discriminative patterns hidden by large-power frequencies. Since each band yields a single kernel, the MKL scheme gathering several kernel representations augmented the class separability.